# Comparative Transcriptome and Co-Expression Network Analyses Reveal the Molecular Mechanism of Calcium-Deficiency-Triggered Tipburn in Chinese Cabbage (*Brassica rapa* L. ssp. *Pekinensis*)

**DOI:** 10.3390/plants11243555

**Published:** 2022-12-16

**Authors:** Shu Zhang, Hanzhong Gao, Lixia Wang, Yihui Zhang, Dandan Zhou, Ali Anwar, Jingjuan Li, Fengde Wang, Cheng Li, Ye Zhang, Jianwei Gao

**Affiliations:** 1Institute of Vegetables, Shandong Key Laboratory of Greenhouse Vegetable Biology, Shandong Branch of National Vegetable Improvement Center, Huanghuai Region Vegetable Scientific Station of Ministry of Agriculture (Shandong), Shandong Academy of Agricultural Sciences, Jinan 250100, China; 2Columbian College of Arts & Sciences, Phillips Hall, The George Washington University, 801 22nd St. NW., Washington, DC 20052, USA; 3College of Life Sciences, Shandong Normal University, Jinan 250061, China; 4College of Life Science, Huangshan University, Huangshan 245061, China

**Keywords:** Chinese cabbage, calcium deficiency, tipburn, RNA-seq, co-expression analysis

## Abstract

Chinese cabbage tipburn is characterized by the formation of necrotic lesions on the margin of leaves, including on the insides of the leafy head. This physiological disorder is associated with a localized calcium deficiency during leaf development. However, little information is available regarding the molecular mechanisms governing Ca-deficiency-triggered tipburn. This study comprehensively analysed the transcriptomic comparison between control and calcium treatments (CK and 0 mM Ca) in Chinese cabbage to determine its molecular mechanism in tipburn. Our analysis identified that the most enriched gene ontology (GO) categories are photosynthesis, thylakoid and cofactor binding. Moreover, the KEGG pathway was most enriched in photosynthesis, carbon metabolism and carbon fixation. We also analyzed the co-expression network by functional categories and identified ten critical hub differentially expressed genes (DEGs) in each gene regulatory network (GRN). These DEGs might involve abiotic stresses, developmental processes, cell wall metabolism, calcium distribution, transcription factors, plant hormone biosynthesis and signal transduction pathways. Under calcium deficiency, CNX1, calmodulin-binding proteins and CMLs family proteins were downregulated compared to CK. In addition, plant hormones such as GA, JA, BR, Auxin and ABA biosynthesis pathways genes were downregulated under calcium treatment. Likewise, *HATs*, *ARLs* and *TCP* transcription factors were reported as inactive under calcium deficiency, and potentially involved in the developmental process. This work explores the specific DEGs’ significantly different expression levels in 0 mM Ca and the control involved in plant hormones, cell wall developments, a light response such as chlorophylls and photosynthesis, transport metabolism and defence mechanism and redox. Our results provide critical evidence of the potential roles of the calcium signal transduction pathway and candidate genes governing Ca-deficiency-triggered tipburn in Chinese cabbage.

## 1. Introduction

Calcium, [Ca^2+^], is a ubiquitous divalent cation in all living systems. Calcium is required for structural roles in the cell wall and membranes as a counter-cation for inorganic and organic anions in the vacuole and intracellular second messenger in the cytosol [1]. Plants growing with adequate calcium in their natural habitats have shoot calcium concentrations between 0.1 and 5% dry weight. As the fifth most abundant element in the earth’s crust, calcium deficiency is rare but may occur in soils with low base saturation and high acidic deposition [2]. Brassica rapa (*Brassica rapa* L. ssp. *Pekinensis*) belongs to the family of Brassicaceae, which is one of the most popular and highly consumed vegetables throughout Eastern Asia. In horticultural crops, especially leafy vegetables such as Chinese cabbage (*Brassica rapa* L. ssp. *Pekinensis*), the [Ca^2+^] that drops below a critical level in fast-growing tissues causes physiological disorders with necrosis at the leaf apex of young developing leaves known as tipburn [3].

The primary cause of tipburn is calcium deficiency, resulting from environmental factors such as greenhouse climate, electrical conductivity (EC), light, CO_2_, or drought stress [4]. However, some causes can also come from the plant, such as poor calcium distribution. Sufficient external calcium ([Ca^2+^]_ext_) is necessary to ensure that the calcium concentration within the cell ([Ca^2+^]_cyt_) remains low. The proper calcium concentration difference on each side of the cell membrane helps build up the cell morphology and suppress the tipburn outbreak.

Calcium homeostasis in plant cells is regulated through various transport elements such as ion exchangers, pumps and channels involved in Ca^2+^ transport. Plant calcium is mainly affected by three main influences: calcium absorption in the root, calcium transfer from root to shoot and competition for cell calcium distribution [5]. Ca^2+^ absorbed from the soil solution through plasma membrane channels is transported from the roots via the xylem to different tissues and organs [6]. However, root Ca^2+^ uptake decreases with increasing distance from the root apex, much higher in apical than basal root zones. Ca transport to aerial plant organs depends on several factors, such as xylem sap Ca^2+^ concentration, balanced mineral nutrition, water uptake and plant water potential, transpiration and growth rate [4,7]. In the plasma membrane of root cells, many calcium-permeable channels have been identified by biochemical and electrophysiological techniques [8,9,10]. Advances in sequencing genomes have led to the identification of different channel families and the functional characterization of specific members. Ca^2+^-permeable channels in root cells were first classified into depolarization-activated channels (DACCs) and hyperpolarization-activated channels (HACCs) by their electrophysiological properties [10]. In the context of signal transduction, there are also the cyclic nucleotide-gated channels (CNGC) family and glutamate receptor homologs (GLRs), which are suggested to account for voltage-independent-channels (VICs)-induced currents [9].

As a critical nutrient, calcium must be taken up and distributed within the plant via the apoplastic and the symplastic pathways. The Casparian strip forms a barrier around endodermal cells to inhibit the uptake of unwanted or toxic substances, which also keeps Ca^2+^ concentrations in the cytosol maintained at the submicromolar level to ensure the ability to generate a Ca^2+^ signature. The Ca^2+^ enters the cytosol of the endodermal cells via channel proteins and is exported into the stellar apoplast via Ca^2+^-ATPases or Ca^2+^/H^+^ antiporters to be loaded into the xylem [11]. When calcium is transported to the shoot and then unloaded and distributed within the leaf cells via calcium influx channels such as CNGC2 [12], a rehearsal of the coding and decoding of the calcium messages occurs.

The availability of contrasting phenotypes for a specific trait offers an excellent opportunity to reveal the genetic variation controlling that trait. Here, we used RNA-seq technology to analyze the transcriptomes of a Chinese cabbage variety under different calcium concentration treatments. We deciphered these data to reveal transcriptome dynamics and transcriptional networks associated with calcium deficiency and identified vital differences that trigger tipburn in Chinese cabbage. In addition, the differential expressed genes (DEGs) and discovery of single nucleotide polymorphisms (SNPs) identified candidate genes that might involve calcium homeostasis in Chinese cabbage. We also performed transcriptome profiles to investigate hormone differences. Thus, this study provides insights into the molecular mechanisms underlying development and the factors that regulate calcium distribution in Chinese cabbage.

## 2. Results

### 2.1. Global Transcriptome Analysis in Chinese Cabbage

The transcriptomic analysis of different treatments of calcium concentration in the Chinese cabbage differing in their calcium deficiency phenotype (Figure 1) can provide crucial systems-level insights into molecular mechanisms underlying plant development and calcium metabolism. To investigate the transcriptome dynamics during the tipburn outbreak, we performed RNA-seq experiments using total RNA isolated from the same stages of plant development and leaves of 21-day-old plants with apparent tipburn phenotype from the Chinese cabbage cultivar Jiaoyan3. All the tissues were analyzed in three independent biological replicates. A total of 357,089,818 high-quality reads (average ~59,514,969 reads from each sample) were generated for each CK and 0 mM Ca treatment from different tissues (Appendix A) and mapped to the Chinese cabbage genome (*Brassica rapa* L. ssp. *Pekinensis*, v3) using TISAT2. The mapped files were processed via StringTie, which generated a consensus transcriptome assembly with 47,615 gene loci, including 47,416 known and 199 novel gene loci. The uniquely mapped reads (47–65 million) for each sample (Appendix A) were processed using Cufflinks to determine the normalized expression level as fragments per kilobase of transcript length per million mapped reads (FPKM) of each transcript. Spearman correlation coefficient (SCC) between the biological replicates of different tissues varied from 0.95 to 0.98, indicating the high quality of the replicates (Figure 2a).

Around 89.53% of genes were identified as being expressed in at least one of the six samples. The number of expressed genes in different samples varied from 92.2% (Ca-1) to 93.8% (Ca-3) in 0 mM Ca and 90.2% (CK-3) to 92.4% (CK-1) in CK (Figure 3a and Appendix A). About 67–73% of genes exhibited very high (FPKM ≥50) expression levels in the different tissues analyzed. The percentage of genes showing high (10 ≤ FPKM ≤ 50) expression was 13–15%. Moreover, the moderate (2 ≤ FPKM ≤ 10) and low (0.1 ≤ FPKM ≤ 2) expressions were 8–11% and 3–6%, respectively (Figure 3b and Appendix A). These analyses showed sufficient transcriptome coverage during the tipburn outbreak in the Chinese cabbage cultivar Jiaoyan3.

### 2.2. Global Comparison of Transcriptomes of Calcium Treatment

To investigate the global differences in the transcriptomic data during a tipburn outbreak in CK and 0 mM Ca treatments, we performed hierarchical clustering and principal component analysis (PCA) based on SCC analysis of average FPKM values for all the expressed genes in at least one of the 16 tissue samples (Figure 2b). These analyses showed a higher correlation of similar physiological stages between the two treatments. Furthermore, as expected, the leaf transcriptome of both treatments clustered together and showed substantial differences with calcium treatment (Figure 2a,b). Therefore, the tissues showing a higher correlation in these analyses are expected to have more similar transcriptomes and functions/activities.

### 2.3. Differential Gene Expression during Tipburn

We identified genes in both the sample groups to investigate the transcriptional differences between CK and 0 mM Ca treatment. We identified 25,919 and 25,060 genes in 0 mM Ca and CK, respectively. A significant portion (41% in both CK and 0 mM Ca) of genes were encoding for transcription factors (TFs) (Figure 3c and Appendix A). Among these, 15,801 TF-encoding genes belonging to 638 families exhibited specific expression in the tipburn outbreak stage. Pkinase, Pkinase_Tyr, Myb_DNA-binding, RRM_1 and P450 TF families were highly represented in the stage. High similarity in the overall transcriptome was detected among different samples of tipburn outbreaks within and across the treatments. However, the number of treatment-specific genes was significantly different. Treatment-specific genes varied from 1562 to 703 for 0 mM Ca and CK, respectively (Figure 4a), while 24,357 genes reported co-expression patterns among treatments, as presented in Figure 4a. Likewise, the heatmap represents the expression of the specific genes in treatments and suggests that the control and Ca treatments were significantly different, as shown in Figure 4b.

The gene ontology (GO) enrichment analyses of all the genes in 0 mM Ca and CK represented genes related to various items. The 165 GO categories belonged to three functional domains: the biological process domain with 85, including 3600 DEGs, the cellular component domain with 24, including 84 DEGs and the molecular function domain with 55, including 2974 DEGs (Appendix A). In the biological process domain, according to padj value (7.91 × 10^−22^), 67 DEGs were significantly enriched to the photosynthesis category, followed by ATP metabolic process category (5.15 × 10^−6^, 61 DEGs) and ribose phosphate metabolic process category (5.15 × 10^−6^, 74 DEGs) (Figure 5a). In the cellular component domain, the DEGs were mainly enriched to the thylakoid category (63 DEGs), followed by the thylakoid part category (63 DEGs) and photosynthetic membrane category (62 DEGs) (Figure 5a). In the molecular function domain, the DEGs were mainly enriched to the cofactor binding category (1.34 × 10^−8^, 185 DEGs), followed by the iron–sulfur cluster binding category (6.46 × 10^−7^, 49 DEGs) and metal cluster binding category (6.46 × 10^−7^, 49 DEGs) (Figure 5a). The most enriched GO categories were generated by BinGo (Figure 4c–e).

The KEGG pathway analysis suggested twenty significantly enriched pathways (Figure 5b and Appendix A). The most enriched pathways were photosynthesis, carbon metabolism, carbon fixation in photosynthetic and biosynthesis of amino acids.

We performed reverse transcription-quantitative polymerase chain reaction (RT-qPCR) analyses for twenty-seven genes showing stage-specific expression in the tipburn tissue samples. The expression pattern of the tested genes revealed by RT-qPCR was similar to those observed in RNA-seq data (Figure 6 and Appendix A), indicating the accuracy of RNA-seq data to reflect the abundance of transcript levels.

### 2.4. Protein—Protein Interaction (PPI) Regulatory Networks Related to Chinese Cabbage Tipburn

We grouped coexpressed gene modules via the functional categories to investigate the gene regulatory network (GRN) during the tipburn in Chinese cabbage. Eight DEG sets were generated; these were associated with the calcium signal pathway, hormone metabolism, cell wall, transport metabolism, developmental process, light responses, abiotic stresses and transcriptional factors (TFs). Using the STRING online database [13] and Cytoscape software [14], a total of 586 DEGs were filtered into the DEGs’ PPI network complex under the stress category, containing 302 nodes and 688 edges (Figure 7a). We used the Cytoscape Hubba plug-in with Maximal Clique Centrality (MCC) method to identify the most significant ten hub genes. The hub gene network contained 10 nodes and 19 edges. These genes were BraA04g021080.3C (member of GST family), BraA05g011900.3C (protein disulfide isomerase), BraA03g047170.3C (putative HSP70), BraA09g026940.3C (glutathione peroxidase), BraA07g036650.3C (endoplasmic reticulum oxidoreductin-1), BraA04g032410.3C (glutathione peroxidase), BraA02g022700.3C (glutathione S-transferase DHAR2-like isoform X2), BraA03g031840.3C (uncharacterized protein, hevein-like preproprotein in *Brassica oleracea*), BraA06g034080.3C (glutathione synthetase), BraA02g022390.3C (glutathione S-transferase U10-like) (Appendix A). The GRN of genes showed a lower CK expression and a higher expression in the 0 mM Ca treatment (Figure 7b). Figure 7c shows the interactive relationships within the top ten hub genes.

In the calcium category, the DEGs PPI network contains 60 nodes and 47 edges (Figure 8a). The top ten hub gene networks contained 10 nodes and 20 edges. BraA08g032770.3C (uncharacterized protein, calreticulin-2 in *Raphanus sativus*) encoded a key protein species in this PPI network and interacted with BraA06g025300.3C (calnexin homolog 1), BraA09g007110.3C (uncharacterized protein, papillar cell-specific calnexin in *Brassica napus*), BraA07g019920.3C (calcium-binding protein CML45), BraA06g024070.3C (uncharacterized protein, calcium-binding protein CML47 in *Brassica napus*), BraA06g019530.3C (calcium-binding protein CML47), BraA03g045230.3C (calcium-binding protein CML41 isoform X1), BraA02g040990.3C (calmodulin-binding protein 60 G-like isoform X1), BraA07g040250.3C (calcium-binding protein CML39) and BraA07g027310.3C (calcium-binding protein CML38-like) (Appendix A). The Ca^2+^-mediated signaling pathway is crucial for environmental adaptation in plants. Calnexin homologs, which act as molecular chaperones located in the endoplasmic reticulum (ER), played an essential role in regulating the cytosolic free calcium concentration ([Ca^2+^]_cyt_) in *Aspergillus nidulans* [15]. The calmodulin-like (CML) protein family appeared to act as a sensor to regulate downstream targets and is not endowed with catalytic activity. CMLs remain poorly characterized at structural and functional levels, even if they are the plants’ largest class of Ca^2+^ sensors. The primary structural theme in CMLs consists of EF-hands, and variations in these domains are predicted to significantly contribute to the functional versatility of CMLs [16]. The GRN of genes showed a lower CK expression and a higher expression in the 0 mM Ca treatment except for BraA03g045230.3C and BraA06g024070.3C (Figure 8b). The BraA08g032770.3C, BraA06g025300.3C and BraA09g007110.3C showed a robust interactive relationship within the top ten hub genes (Figure 8c).

The cell wall category contains 95 nodes and 69 edges in this research. The top ten hub gene networks contained 10 nodes and 31 edges (Figure 9a). BraA01g037350.3C encode a pectinesterase and interact with BraA05g032750.3C (pectinesterase), BraA03g001520.3C (pectate lyase 5), BraA09g001840.3C (pectinesterase), BraAnng003600.3C (pectinesterase), BraA05g000370.3C (pectinesterase), BraA09g003200.3C (pectate lyase 11), BraA04g008830.3C (pectate lyase 14), BraA08g034730.3C (pectate lyase 1) and BraA07g009090.3C (pectate lyase 9 isoform X1) (Appendix A). The GRN of genes showed a lower CK expression and a higher expression in the 0 mM Ca treatment except for BraA07g009090.3C (Figure 9b). The BraA01g037350.3C and BraA05g032750.3C showed a robust interactive relationship within the top ten hub genes (Figure 9c).

During the Chinese cabbage tipburn outbreak stage, the developmental associated factors also varied. In this research, the development category contained 94 nodes and 37 edges. The top ten hub gene networks contained 10 nodes and 7 edges (Figure 10a). The BraA01g019700.3C (homeobox-leucine zipper protein HAT4 in *Brassica napus*), BraA07g037840.3C (transcription factor BEE 3-like in *Brassica napus*), BraA07g023360.3C (nuclear transcription factor Y subunit C-2), BraA01g034360.3C (transcription factor TCP18-like isoform X1), BraA09g022870.3C (nuclear transcription factor Y subunit B-2), BraA02g021930.3C (transcription factor BEE 3-like), BraA04g030060.3C (transcription factor PAR1-like), BraA03g022440.3C (transcription factor PAR1-like in *Brassica napus*), BraA02g023490.3C (protein CUP-SHAPED COTYLEDON 3-like) and BraA03g018340.3C (protein HAIKU1-like isoform X1) constituted the hub gene network (Appendix A). The GRN of genes showed a lower CK expression and a higher expression in the 0 mM Ca treatment (Figure 10b). The BraA01g019700.3C showed a robust interactive relationship with other hub genes (Figure 10c).

The plant hormone signal also activated the regulatory pathway responding to calcium-deficiency-induced tipburn. The hormone category contained 105 nodes and 216 edges. The top ten hub gene networks contained 10 nodes and 17 edges (Figure 11a). The BraA10g025540.3C (protein TIFY 9-like isoform X1 in *Brassica napus*), BraA09g034900.3C (protein TIFY 5A-like), BraA08g023500.3C (protein TIFY 5A-like isoform X2 in *Brassica napus*), BraA02g020000.3C (protein TIFY 7 isoform X1 in *Brassica napus*), BraA07g029850.3C (protein TIFY 7 isoform X1 in *Brassica napus*), BraA05g000390.3C (ethylene response sensor 1), BraA01g030980.3C (ethylene receptor 2), BraA10g000500.3C (1-aminocyclopropane-1-carboxylate synthase 2 in *Raphanus sativus*), BraA09g051070.3C (1-aminocyclopropane-1-carboxylate synthase-like protein 1 in *Eutrema salsugineum*) and BraA09g056530.3C (1-aminocyclopropane-1-carboxylate oxidase 1) constituted the hub gene network (Appendix A). The GRN of genes showed a lower CK expression and a higher expression in the 0 mM Ca treatment (Figure 11b). The BraA10g025540.3C, BraA09g034900.3C and BraA08g023500.3C showed a robust interactive relationship with other hub genes (Figure 11c).

Light is one of the main signals perceived by plants that affect plant growth, development and function [17]. In this research, the light category contained 79 nodes and 685 edges. The top ten hub gene networks contained 10 nodes and 45 edges (Figure 12a). The BraA09g033700.3C (photosystem I reaction center subunit III in *Brassica napus*), BraA07g010120.3C (photosystem I reaction center subunit III), BraA06g000430.3C (photosystem I reaction center subunit V), BraA05g037360.3C (chlorophyll a-b binding protein CP29.2 in *Brassica napus*), BraA09g045720.3C (chlorophyll a-b binding protein 6 in *Brassica napus*), BraA07g021900.3C (chlorophyll a-b binding protein 6 in *Brassica oleracea var. oleracea*), BraA09g062800.3C (photosystem I subunit O in *Brassica napus*), BraA10g012410.3C (chlorophyll a-b binding protein 3 in *Brassica napus*), BraA09g016320.3C (photosystem I chlorophyll a-b-binding protein 3-1) and BraA01g029170.3C (photosystem I chlorophyll a-b-binding protein 3-1) constituted the hub gene network (Appendix A). The GRN of genes showed a lower expression in the 0 mM Ca treatment and a higher expression in CK (Figure 12b). In this study, the photosystem-related genes BraA09g033700.3C, BraA07g010120.3C and BraA06g000430.3C showed a robust interactive relationship with other hub genes (Figure 12c).

The transcription factor (TF) category contained 22 nodes and 14 edges (the core interaction network) in this study. The top ten hub gene networks contained 10 nodes and 4 edges (Figure 13a) including BraA02g012610.3C (transcription factor bHLH041), BraA02g040390.3C (basic leucine zipper 63-like), BraA03g024170.3C (transcription factor CPC in *Brassica oleracea var. oleracea*), BraA06g035500.3C (transcription factor bHLH78 isoform X1), BraA01g042000.3C (transcription factor MYB108), BraA10g024430.3C (transcription factor WER isoform X1 in *Brassica napus*), BraA03g024500.3C (transcription factor MYB12-like), BraA04g028280.3C (transcription factor bHLH51-like), BraA09g048650.3C (basic leucine zipper 61) and BraA09g038670.3C (dof zinc finger protein DOF1.3-like in *Brassica napus*) (Appendix A). The GRN of genes showed a lower CK expression and a higher expression in the 0 mM Ca treatment, except BraA02g012610.3C and BraA02g040390.3C (Figure 13b). The BraA02g012610.3C and BraA02g040390.3C showed a robust interactive relationship with other hub genes (Figure 13c).

In this research, the transporter category contained 61 nodes and 96 edges. The top ten hub gene networks contained 10 nodes and 45 edges (Figure 14a). The BraA03g063280.3C (aspartate aminotransferase, cytoplasmic isozyme 2 isoform X1), BraA03g004710.3C (aspartate aminotransferase 3 in *Tarenaya hassleriana*), BraA10g020790.3C (aspartate aminotransferase, cytoplasmic isozyme 1 isoform X2 in *Tarenaya hassleriana*), BraA03g002960.3C (glutamate dehydrogenase 2-like in *Brassica napus*), BraA04g022190.3C (aspartate aminotransferase, mitochondrial in *Tarenaya hassleriana*), BraA02g006550.3C (glutamine synthetase cytosolic isozyme 1-4-like in *Raphanus sativus*), BraA03g038160.3C (glutamine synthetase cytosolic isozyme 1-3-like in *Brassica napus*), BraA09g023350.3C (glutamate decarboxylase 4-like in *Brassica napus*), BraA02g035300.3C (glutamate decarboxylase 4-like in *Raphanus sativus*) and BraA07g031850.3C (glutamate decarboxylase 2-like isoform X1 in *Raphanus sativus*) constituted the hub gene network (Appendix A). All of the genes of GRN showed a lower expression in CK and a higher expression in the 0 mM Ca treatment (Figure 14b). The BraA03g063280.3C, BraA03g004710.3C, BraA10g020790.3C, BraA03g002960.3C and BraA04g022190.3C showed a solid interactive relationship with other hub genes (Figure 14c).

### 2.5. Metabolome Analysis to Identify the Hormones Related to Chinese Cabbage Tipburn

To compare the content of hormones in Chinese cabbage during the tipburn outbreak, we analyzed the leaf samples of 0Ca and CK by LC-MS/MS. Eighty-eight hormones were detected in the two varieties, including 2 ABA, 26 Auxin, 36 CK, 1 ETH, 10 GA, 9 JA, 2 SA and 2 SL (Appendix A). The hormones were enriched in four KEGG categories: plant hormone signal transduction, metabolic pathways, diterpenoid biosynthesis and biosynthesis of secondary metabolites (Figure 15a). After unit variance scaling, the hormones were clustered into eight classes (Figure 15b). Salicylic acid (SA) is required for local and systemic disease resistance responses in higher plants, and SA-dependent cell death has been reported to correlate closely with Ca^2+^ homeostasis. In this research, salicylic acid (SA) and SA β-glucoside (SAG) also accumulated in response to calcium deficiency conditions. Previous studies have shown that Trp can promote plants’ growth and development and improve tolerance to environmental stresses such as drought and salinity [18]. The most abundant hormone during tipburn is salicylic acid 2-O-β-glucoside (SAG), followed by L-tryptophan (TRP) and N6-Isopentenyl-adenine-7-glucoside (iP7G). Furthermore, 11 hormones significantly differ between 0Ca and CK during tipburn (Figure 15c).

## 3. Discussion

The molecular mechanisms underlying calcium-deficiency-triggered Chinese cabbage tipburn are poorly understood. Therefore, we used the RNA-seq approach to detect the transcriptome dynamics in the tipburn outbreak stages and investigated the potential molecular mechanism (Appendix A). More than 90.2% of the Chinese cabbage genes were expressed in at least one biological replicate of the CK and 0 mM Ca treatment samples. Thus, RNA-seq facilitated the discovery of novel genes and their expression pattern. The expression data across the different Ca concentration treatments showed high reproducibility, and each treatment was clearly distinguished in the PCA plot, suggesting that significant changes in gene expression occur from 0 mM Ca treatment to CK. The transcriptome analyses with the inference of co-expression networks and transcriptional categories identified several coregulated and specific transcriptional programs within and across the treatment associated with calcium deficiency and stress adaptation.

The GO enrichment analyses revealed prolonged photosynthesis and thylakoid activity with higher expression of genes involved in the photosystem in 0 mM Ca treatment samples. Tatiani et al. [19] found that leaf photosynthesis positively correlated with leaf Ca concentration because calcium deficiency affects the photosynthetic process, causing decreased carboxylation efficiency and photosynthetic capacity. Rangnekar et al. [20] also found that after 8–10 days of calcium deprivation, chlorophyll (A + B) was reduced by 57% and the organic acids and lipids were considerably decreased (50% or more), and these reductions indicate that a restriction in the movement or utilization of early products of photosynthesis outside the chloroplast is one of the possible effects of calcium starvation. Several chloroplast proteins involved in functionally diverse processes bind Ca^2+^ and are regulated by Ca^2+^ in vitro [21,22]. The import of Ca^2+^ across the thylakoid membrane depends on a light- or ATP-induced transthylakoid proton gradient. This might be why calcium deficiency downregulated the genes that facilitate chlorophyll metabolism and photosystem, thus leading to photosynthesis abnormality.

The glutathione can reinforce the circadian clock to gate the immune response by enhancing *TOC1* expression [23]. Moreover, deficient glutathione in guard cells facilitates abscisic acid-induced stomatal closure, affecting ROS production’s function downstream in the ABA signaling cascade [24]. There is substantial evidence that HSPs play critical physiological roles in normal conditions and situations involving systemic and cellular stress. Protein disulfide isomerase (PDI) is an essential multifunctional protease and widely exists in eukaryotes, and is also known as protein thiol-oxidoreductase; it has the activities of oxidase, isomerase and disulfide reductase and can also perform a molecular chaperone function to the endoplasmic reticulum (ER) stress in plants [25].

Cell wall metabolism is essential to basal stress and disease responses [26], and calcium deficiency can affect cell wall structure components. Pectinesterases (PMEs) are vital regulators in catalyzing the pectin to form and modify cell walls via the demethylesterification of cell wall pectin [27,28]. Pectate lyases (PLs) and other cell wall degrading enzymes can act as virulence factors, which enhance the potential to cause disease and play an essential role in the infection process.

Nuclear transcription factor Y (NF-Y) could directly bind to the cis-acting elements in the cytochrome P450 3A subfamily (CYP3A) proximal promoters, which play a vital role in the metabolism of endogenous chemicals and xenobiotics [29]. Teosinte branched1, Cycloidea, Proliferating cell factor (TCP) family proteins are plant-specific transcription factors (TFs) and play an essential role in many plant biological processes, especially in the regulation of leaf curvature [30]. HAIKU1 (IKU1, also known as AtVQ14) plays an essential role in endosperm growth and determining *Arabidopsis* seed size through the IKU pathway [31,32].

Many TF families have been implicated in stress adaptation [33]. Among the families of transcription factors related to plant stress resistance, there are four significant categories of bZIP, WRKY, AP2/EREBP and MYB [34]. The bHLH family is the second-largest plant family after the MYB family [35]. The bHLH transcription factor regulates carpel, anther, epidermal cell and stomatal development [36]. As shown in Figure 11, numerous plant hormone biosynthesis genes were upregulated in the control but downregulated in Ca. These findings suggest that the Ca signaling pathway strongly correlates with plant hormone biosynthesis, facilitating tipburn and inducing stress tolerance. In addition, bHLHs regulate metabolic processes, including the biosynthesis of alkaloids and nicotine [37]. Many bHLHs play a crucial role in light signal regulation and are also involved in the function of signaling hormones such as abscisic acid (ABA), brassinosteroids (BRs), ethylene, gibberellin and jasmonic acid [38,39,40]. In addition, bHLH transcription factors are involved in plant responses to stress, such as hypothermia, drought, salinity and iron-deficient abiotic stress [41]. MYB transcription factors play an important role in abiotic stress responses [42]. Plants use stomata on the epidermis to exchange carbon dioxide and water with the atmosphere. Regulation of the stomatal aperture is one of the most critical ways plants control water loss [43]. MYB60 is the first transcription factor shown to participate in the regulation of stomatal movement [44]. In addition, MYBs can rely on ABA signals to regulate plant stomatal movement [45,46,47,48]. During the tipburn outbreak stage, the Chinese cabbage appeared with dry edges on the young leaves. The TFs might adjust stomatal movement to prevent water loss and help Chinese cabbage accommodate the Ca-deficiency situation.

In Chinese cabbage, the transcriptome-wide identification and characterization of circular RNAs in leaves in response to calcium-deficiency-induced tip-burn has been built [49]. Furthermore, discovering transcriptional modules can identify GRNs that control biological processes associated with biological traits [50,51,52,53]. Therefore, we constructed the transcriptional modules linking TFs with their potential binding motifs and coexpressed target genes for the two crucial treatments (CK and 0mM Ca) of calcium concentration to determine the differential Ca-related physiological phenotype. The primary cause of tipburn is often considered to be Ca^2+^ deficiency. Ca^2+^ is a component of cell walls and maintains cell function as a messenger signal, so Ca^2+^ deficiency causes cell death or necrosis because of abnormal cell formation. Internal browning of cabbage and blossom end rot of tomato fruit are also due to Ca^2+^ deficiency in the same way as tipburn. In our research, the cell wall construction ingredients, pectinesterase and pectate lyases, act as essential elements in 0 mM Ca samples. PMEs are vital regulators in catalyzing pectin to form pectate. Moreover, plant PMEs have processive activities and create polyanionic stretches in a high degree of methylesterification (HG), forming cooperative Ca^2+^ crosslinks and forming a stiff gel [54]. The stiff gel material might be one of the reasons for the crispy dry edge and make the leaves lose their flexibility.

Chinese cabbage tipburn always develops with pathogen infections, further reducing production [55,56]. As part of the stress response, plants produce glutathione (GSH). GSH acts as an antioxidant by quenching reactive oxygen species and is involved in the ascorbate–glutathione cycle that eliminates damaging peroxides [57]. In this research, the glutathione synthetase (GS), glutathione transferases (GSTs), as well as glutathione peroxidase (GP) showed the importance in the GRN. GSH is the most abundant antioxidant and a primary detoxification agent in cells. It is synthesized through a two-enzyme reaction catalyzed by glutamate-cysteine ligase (GSL) and GS, and its level is well regulated in response to redox change. Evidence suggests that GSH may play essential roles in cell signaling [58]. The activity and expression of GSTs depend on several less-known endogenous and well-described exogenous factors, such as the developmental stage, presence and intensity of different stressors [59]. The functional studies revealed that overexpression or silencing of specific GSTs could markedly modify disease symptoms and pathogen multiplication rates [60]. Interestingly, in this study, the GSL did not appear in the core GRN, implying other mechanisms might be behind it.

Many phytohormones, including abscisic acid, brassinosteroids and salicylic acid, play vital roles in the ability of plants to respond to abiotic and biotic stress [61]. This research found that calcium deficiency triggered DEGs’ core GRN involved in the jasmonate acid (JA), IAA, ethylene, abscisic acid (ABA) and gibberellin (GA) pathway. Previous research found that ABA accumulated during the Ca^2+^ deficiency stress, which suggested ABA have a regulatory role in response to Ca^2+^ deficiency-induced tipburn in *Brassica*. *rapa* [49]. Ethylene interacts with nutrient uptake and controls plant responses under growth-limiting conditions or stress. Since the tipburn outbreak usually copes with the overdose of nitrate nutrition, ethylene can also regulate the expression of essential nitrate transporters, namely NRT1.1 and NRT2.1. Under high nitrate concentrations, there is an ethylene-dependent downregulation of the high-affinity nitrate transporter NRT2.1, while the nitrate transceptor NRT1.1 is upregulated [62]. These results suggest ethylene DEGs might participate in tipburn through nitrate pathways. Moreover, there was evidence indicating GA and ABA crosstalk in the coordination of Ca^2+^-ATPase, Ca^2+^/H^+^ exchangers (CAX) activity in the tonoplast and blossom end rot (BER) development symptoms. Moreover, inhibitors of GA biosynthesis may affect the metabolism of other hormones, such as increasing levels of cytokinins and ABA, and adversely affect ethylene biosynthesis. TIFY is a jasmonic acid (JA) signaling protein associated with defence responses against pathogens [63,64]. Ethylene can mediate crosstalk between calcium-dependent protein kinase and MAPK signaling that controls plant stress responses [65]. The ethylene response is negatively regulated by a family of five ethylene receptor genes in *Arabidopsis* [66]. 1-aminocyclopropane-1-carboxylate synthase can participate in transcription via WRKY33 and 1-aminocyclopropane-1-carboxylate oxidase by modulating messenger RNA stability to induce ethylene synthesis during stress [67]. Therefore, further analysis of the DEGs can help us uncover each hormone’s role in the processes that regulate the development of Ca^2+^ deficiency tipburn symptoms.

In order to sense and respond to these fluctuating conditions, higher plants possess several families of photoreceptors that can monitor light from UV-B to the near-infrared (far-red). The light sensors are also the signal input resources of the circadian clock system that allow plants to anticipate and prepare for daily and seasonal changes in surrounding environments [68]. Photosystem I (PSI) is a large protein supercomplex that catalyzes the light-dependent oxidation of plastocyanin (or cytochrome c6) and the reduction of ferredoxin. This catalytic reaction is realized by a transmembrane electron transfer chain consisting of a primary electron donor (a special chlorophyll (Chl) pair) and electron acceptors A0, A1, and three Fe4S4 clusters, FX, FA and FB [69]. Light capture, the first and most important event during this process, is mediated by light-harvesting chlorophyll a/b-binding (Lhc) proteins. These proteins comprise a plant-specific superfamily (known as Lhc) that is characterized by the presence of a chlorophyll-binding (CB) domain in the transmembrane alpha helix [70,71,72].

Under calcium-deficient conditions, plants induce genes involved in iron uptake and translocation. This response to calcium deficiency is regulated by transcriptional networks mediated by transcription factors (TFs) and protein-level modification of critical factors by ubiquitin ligases [73]. The basic helix–loop–helix (bHLH) superfamily, in which proteins are highly conserved, is the second-largest transcription factor (TF) family across eukaryotic kingdoms after the MYB superfamily [74,75,76].

Cytosolic Ca^2+^ ([Ca^2+^]_cyt_) levels increase in plant cells in response to abiotic stress. With this increase, calcium-interacting proteins simultaneously activate several mechanisms, such as Ca^2+^-dependent protein kinases, calmodulin, calmodulin-related proteins, calcineurin-like proteins and calcium-binding EF-hand proteins [77,78]. Aspartate aminotransferase (AAT) is a crucial enzyme in synthesizing amino acids. It is essential in regulating almost all organisms’ carbon and nitrogen metabolism [79]. Glutamine synthetase (GS) is the major assimilatory enzyme for ammonia produced from N fixation and nitrate or ammonia nutrition. It also reassimilates ammonia released due to photorespiration and the breakdown of proteins and nitrogen transport compounds. γ-aminobutyric acid (GABA) synthesis from glutamate is catalyzed by cytosolic glutamate decarboxylase (GAD; EC 4.1.1.15); GABA is then oxidized in mitochondria via two successive steps, catalyzed by GABA transaminase and succinic semialdehyde dehydrogenase (SSADH; EC 1.2.1.16), to form succinate, which then enters the tricarboxylic acid cycle [80].

The previous research showed salicylic acid (SA) accumulated in response to conditions of calcium deficiency, and both total SA and SA β-glucoside (SAG) in tipburn-susceptible plants were ~3-fold higher than in resistant plants following Ca^2+^ deficiency treatment [55]. Furthermore, SA-dependent cell death has been reported to correlate closely with Ca^2+^ homeostasis [81]. In our study, the SAG expression level was ~2-fold higher than CK, suggesting the solid Hoagland medium is a reliable system for calcium-deficiency-triggered Chinese cabbage tipburn. In addition, tryptophan (Trp) is an essential amino acid for protein synthesis. In animals and plants, Trp is a precursor of melatonin (*N*-acetyl-5-methoxytryptamine), which contributes to versatile physiological roles in plant growth, development and protection against biotic and abiotic stresses [82,83]. In this research, the Trp level in 0Ca was significantly higher than in CK, implying that Trp is critical for calcium deficiency accommodation. Moreover, the 1-Aminocyclopropanecarboxylic acid (ACC), which belongs to the ethylene family, also showed a dramatic high in 0Ca compared to CK. Furthermore, because the ACC is a precursor of ethylene biosynthesis and is always connected with Fe absorption, calcium deficiency might cause another iron uptake abnormality [84]. These results, matched with the GRN analysis, suggested the reliability of the bioinformatic assistant regulatory networks analysis. Interestingly, the JA total content in the metabolomic analysis was much lower than SAG, Trp and ACC, suggesting that JA is a possible micro-efficiency hormone in tipburn resistance.

The PPI network based on the STRING database can quickly extract the predicted association of a particular group of proteins. Nevertheless, this strategy relies on the database’s accuracy and refreshing frequency. The relationship between the proteins is also based on the network drawing algorithm. In addition, we still need a wet experiment to confirm the predicted interactions.

In conclusion, RNA-seq data generated from calcium deficiency material in this research present a robust resource to study Chinese cabbage tipburn. We identified gene sets, determined their enriched biological processes (pathways) and defined coexpressed gene sets with high temporal resolution. This study also suggests that transcriptional profiles, deduced GRNs and molecular genetic approaches can help identify the most promising candidate genes and establish their role in Chinese cabbage tipburn.

## 4. Material and Methods

### 4.1. Plant Material and Sampling

The Chinese cabbage variety Jiaoyan3 seeds, provided by Associate Prof. Lingqiang Sun, Qingdao Agricultural and Rural Bureau, China. The Chinese cabbage variety Jiaoyan3 seeds were surface sterilized with 1% sodium hypochlorite in a vertical flow clean bench for four minutes plus two times and then washed with sterilized distilled water five times to remove the excessive sodium hypochlorite. Furthermore, Chinese cabbage seeds were transferred to Hoagland solid medium in a deep plastic box with or without Ca^2+^. In order to chelate the excessive Ca^2+^ in the medium, 150 mM Ethylenebis (oxyethylenenitrilo) tetra (acetic acid) (EGTA, sigma) were added into the 0 mM Ca Hoagland solid medium [85,86]. Two groups of Chinese cabbage (*Brassica rapa* L. ssp. *Pekinensis*) of different calcium treatments with contrasting phenotypes for tipburn, CK (control) and 0 mM Ca (tipburn), were used in this study. Chinese cabbage plants were grown in the climate chamber at 20 ± 2 °C, relative humidity of 65% with a 14 h light and 10 h dark photoperiod. The fluence rate of white light was ~110 μmol m^−2^ s^−1^. Leaves without (or with) significant calcium deficiency phenotype were collected in three biological replicates on the 21st day after emergence, representing CK-1, CK-2, CK-3 and Ca-1, Ca-2 and Ca-3, respectively. For RNA extraction, the tissue samples were collected on dry ice and snap-frozen in liquid nitrogen.

### 4.2. Illumina Sequencing, Read Mapping and Differential Gene Expression Analyses

For RNA-seq, total RNA extraction and library preparation for each sample was performed by the Novogene company standard protocol. All six libraries (2 samples in three biological replicates) were sequenced on the Illumina Hiseq platform (HiSeq 2000) to generate 125 bp/150 bp paired-end reads. Raw data (raw reads) of fastq format were first processed through in-house Perl scripts. This step obtained clean data (clean reads) by removing reads containing adapter, ploy-N and low-quality reads from raw data. At the same time, Q20, Q30 and GC content of the clean data were calculated. All the downstream analyses were based on clean data with high quality. The high-quality filtered reads were mapped to the Chinese cabbage genome using Hisat2 v2.0.5. The mapped output was processed via Cufflinks (v2.0.2) to obtain FPKM for all the Chinese cabbage genes in each sample. Correlation between the biological replicates was determined via calculating SCC. Hierarchical clustering and PCA analysis were performed using corrplot and prcomp utilities in the R package. Cuffdiff determined the differential expression between different samples. The genes exhibiting a difference of at least twofold change with corrected *p*-value after adjusting with false discovery rate (*q*-value) ≤ 0.05 were considered to be significantly differentially expressed. The SS scoring algorithm identified both cultivars’ stage-specific/preferential genes [87]. The higher value of the SS score of a gene in a particular stage signifies its more specific expression at that stage. Row-wise, Z-scores were determined for a given set of genes, and heatmaps were generated using heatmap2 of the R package.

### 4.3. GO KEGG and Pathway Enrichment Analysis

Gene ontology enrichment analysis for differentially expressed gene sets was performed using Cytoscape (BiNGO plug-in). The *p*-value for enrichment was calculated for each represented GO term and corrected via the Benjamini–Hochberg error correction method. The GO terms exhibiting a corrected (after adjusting with false discovery rate) *p*-value of ≤0.05 were considered significantly enriched. In addition, KEGG pathway enrichment was performed using a cluster profile R package to test the statistical enrichment of differential expression genes. In addition, identified metabolites were annotated using the KEGG compound database (http://www.kegg.jp/kegg/compound/, accessed on 1 August 2021), annotated metabolites were then mapped to the KEGG pathway database (http://www.kegg.jp/kegg/pathway.html, accessed on 1 August 2021). Pathways with significantly regulated metabolites mapped were then fed into MSEA (metabolite sets enrichment analysis), and the hypergeometric test’s *p*-values determined their significance.

### 4.4. Real-Time Quantitative PCR Analysis

The results of RNA-seq were validated via Real-time PCR (RT-qPCR) experiments. Total RNA was extracted from each sample using Trizol reagent (Invitrogen, Carlsbad, CA, USA) and treated with Rnase-free Dnase I (TaKaRa, Dalian, China) for 45 min according to the manufacturer’s protocol. First-strand cDNA was synthesized from 1 µg of total RNA using a PrimeScript 1st Strand cDNA Synthesis Kit (TaKaRa). Then, RT-qPCR was carried out using SYBR Green Master Mix (TaKaRa) by IQ5 Real-Time PCR Detection System (Bio-Rad, Hercules, CA, USA). The gene-specific primers designed using Primer Express (v3.0) software for each gene are listed in Appendix A). The real-time PCR analysis was performed using three biological replicates for each tissue sample and three technical replicates of each biological replicate. The actin gene was used as a constitutive expression control in the RT-qPCR experiments. The PCR cycling conditions comprised an initial polymerase activation step of 95 °C for 1 min, followed by 40 cycles of 95 °C for 10 s and 60 °C for 30 s. After each PCR run, a dissociation curve was designed to confirm the product’s specificity and avoid primer dimers’ production. The relative amounts of the amplification products were calculated by the comparative 2^−∆∆CT^ method.

### 4.5. Protein—Protein Interaction Analysis

PPI network can help us identify the essential genes and essential gene modules involved in tipburn from the interaction level. The PPI networks came from the different categories of RNA-seq DEGs. First, PPI information of DEGs was acquired from the Search Tool for the Retrieval of Interacting Genes (STRING) database (http://www.string-db.org/, accessed on 1 August 2021). Then, Cytoscape software was used for the construction of the PPI network. At last, the hub gene analysis was carried out by Cytoscape Hubba plug-in, and the top ten nodes were ranked by the Maximal Clique Centrality (MCC) method with default parameters.

### 4.6. Hormone Identification and Quantification for Metabolomic Analysis

The fresh plant sample was harvested, frozen in liquid nitrogen, ground into powder and stored at −80 °C. Then prepared for further LC-MS/MS analysis as described previously [88,89]. The internal standards are shown in (Appendix A).

The sample extracts were analyzed using a UPLC-ESI-MS/MS system (UPLC, ExionLC™ AD, https://sciex.com.cn/ (accessed on 1 August 2021); MS, Applied Biosystems 6500 Triple Quadrupole, https://sciex.com.cn/, accessed on 1 August 2021). The analytical conditions followed the previous methods [90,91,92].

Linear ion trap (LIT) and triple quadrupole (QQQ) scans were acquired on a triple quadrupole-linear ion trap mass spectrometer (QTRAP), QTRAP^®^ 6500+ LC-MS/MS System, equipped with an ESI Turbo Ion Spray interface, operating in both positive and negative ion mode and controlled by Analyst 1.6.3 software (Sciex). The ESI source operation parameters were as follows: an ion source, ESI+/−; source temperature 550 °C ion spray voltage (IS) 5500 V (Positive), −4500 V (Negative); curtain gas (CUR) was set at 35 psi, respectively.

Phytohormones’ contents were detected by MetWare (http://www.metware.cn/, accessed on 1 August 2021) based on the AB Sciex QTRAP 6500 LC-MS/MS platform.

Unsupervised PCA (principal component analysis) was performed by statistics function prcomp within R (www.r-project.org, accessed on 1 August 2021). The data were unit variance scaled before unsupervised PCA. The HCA (hierarchical cluster analysis) results of samples and metabolites were presented as heatmaps with dendrograms, while Pearson correlation coefficients (PCC) between samples were calculated by the cor function in R and presented as only heatmaps. Both HCA and PCC were carried out by R package heatmap. Absolute Log2FC (fold change) determined significantly regulated metabolites between groups.

## Figures and Tables

**Figure 1 plants-11-03555-f001:**
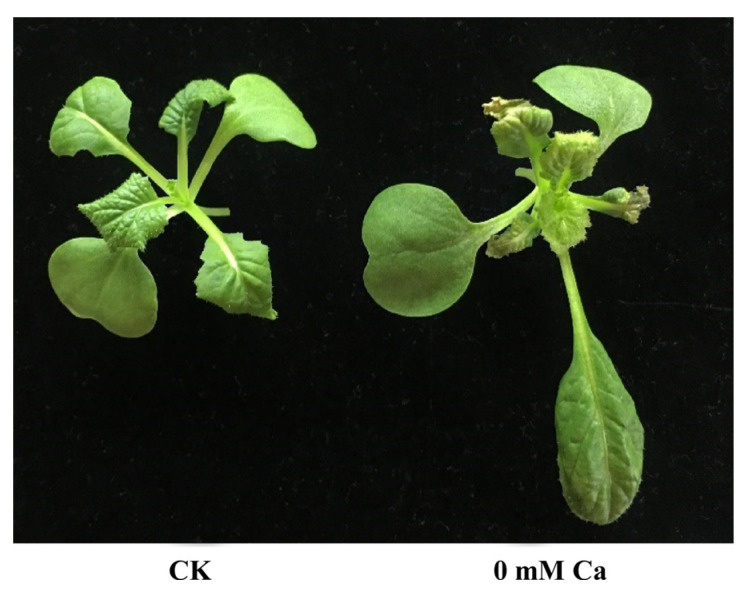
Phenotype comparison of Chinese cabbage. CK indicates Chinese cabbage grown on standard Hoagland medium, 0 mM Ca stands for Chinese cabbage grown on Hoagland medium (without Ca^2+^). The samples were collected on the 21st day after germination.

**Figure 2 plants-11-03555-f002:**
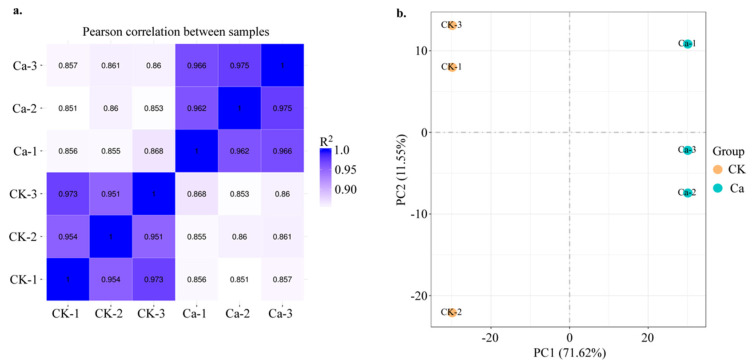
Pearson correlation and PCA analysis. (**a**) Pearson correlation between samples. (**b**) Principle component analysis between samples.

**Figure 3 plants-11-03555-f003:**
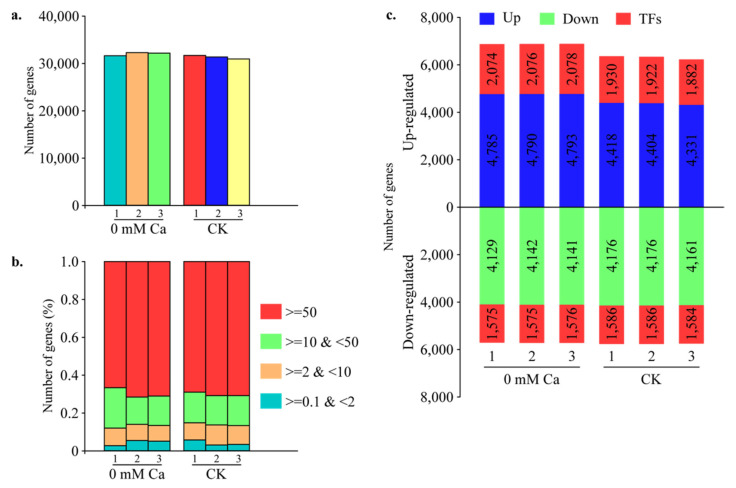
Gene expression in CK and 0 mM Ca treatment Chinese cabbage seedlings. (**a**) A total number of genes expressed in CK and 0 mM Ca treatment. (**b**) Fraction of genes expressed at different samples (based on FPKM). (**c**) Comparison of genes in up and downregulation in CK and 0 mM Ca treatment.

**Figure 4 plants-11-03555-f004:**
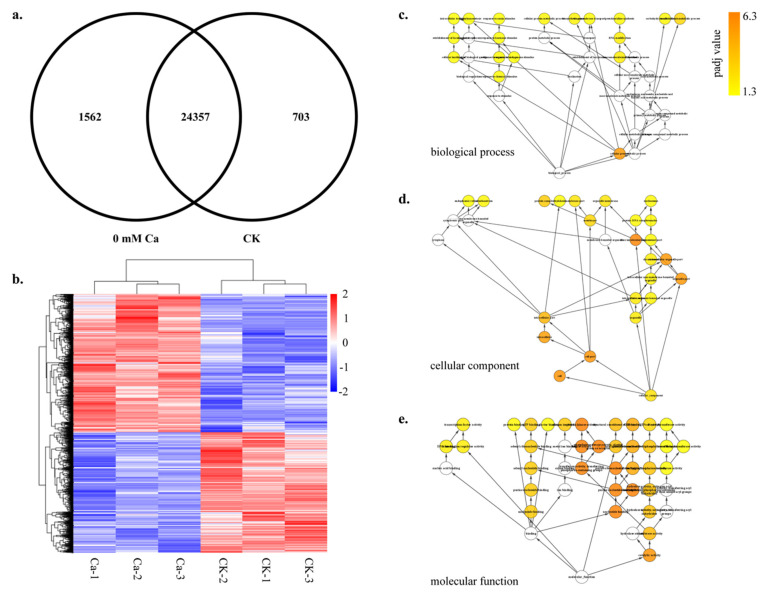
Preferential expression of genes during tipburn outbreak in Chinese cabbage. (**a**) Venn diagram of identified DEGs between samples in Chinese cabbage during calcium-deficiency-induced tipburn. (**b**) Heatmap showing the expression profile of preferentially expressed genes in CK and 0 mM Ca treatment samples. (**c**–**e**) Gene ontology (GO) enrichment of preferentially expressed genes in CK and 0 mM Ca treatment samples. Colors are shaded according to the significance level (white means no significant difference; color scale, yellow *p*-value = 0.05, orange *p*-value < 0.0000005).

**Figure 5 plants-11-03555-f005:**
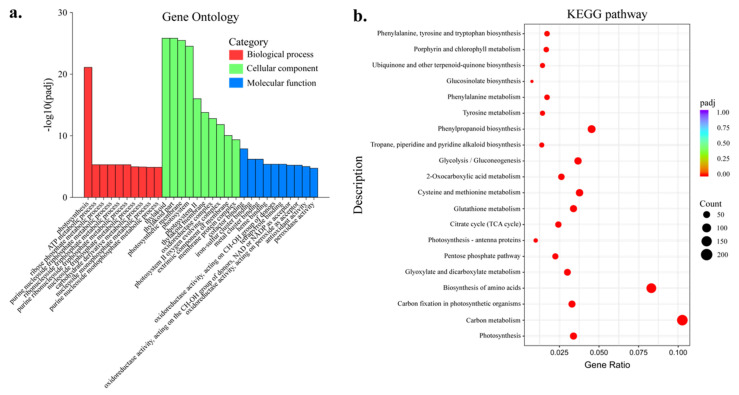
GO and KEGG analysis. (**a**) GO classification of the DEGs. (**b**) KEGG enrichment of the DEGs.

**Figure 6 plants-11-03555-f006:**
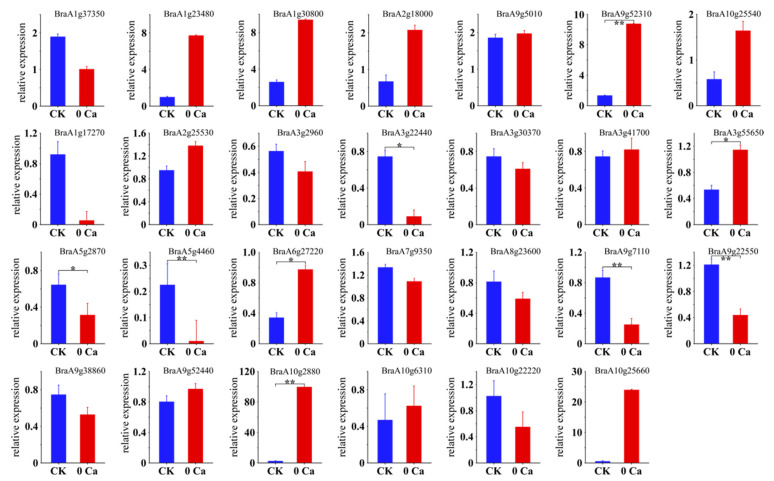
RT-qPCR verification of the DEGs. Asterisks indicate statistical significance using Student’s *t*-test: *, *p* ≤ 0.05, **, *p* ≤ 0.01.

**Figure 7 plants-11-03555-f007:**
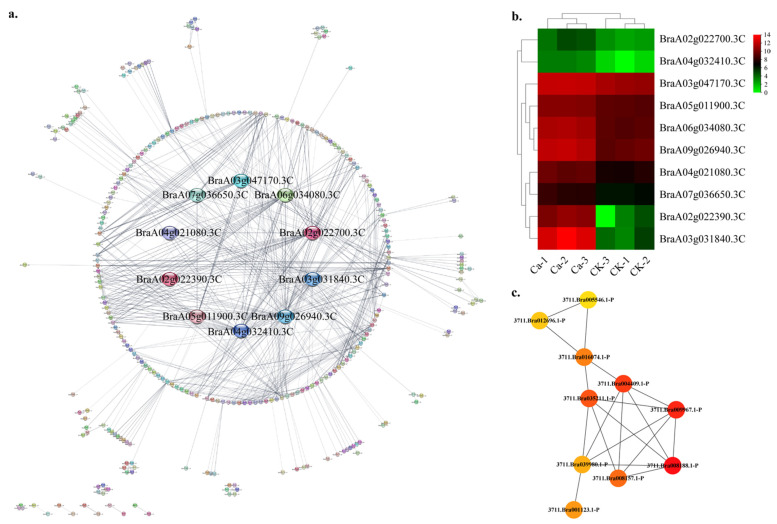
Protein–protein interaction (PPI) analysis in stress-related DEGs category. (**a**) PPI network in stress category. (**b**) Heatmap of the hub gene expression pattern. (**c**) Hub genes’ interactive relationship of the gene regulatory network.

**Figure 8 plants-11-03555-f008:**
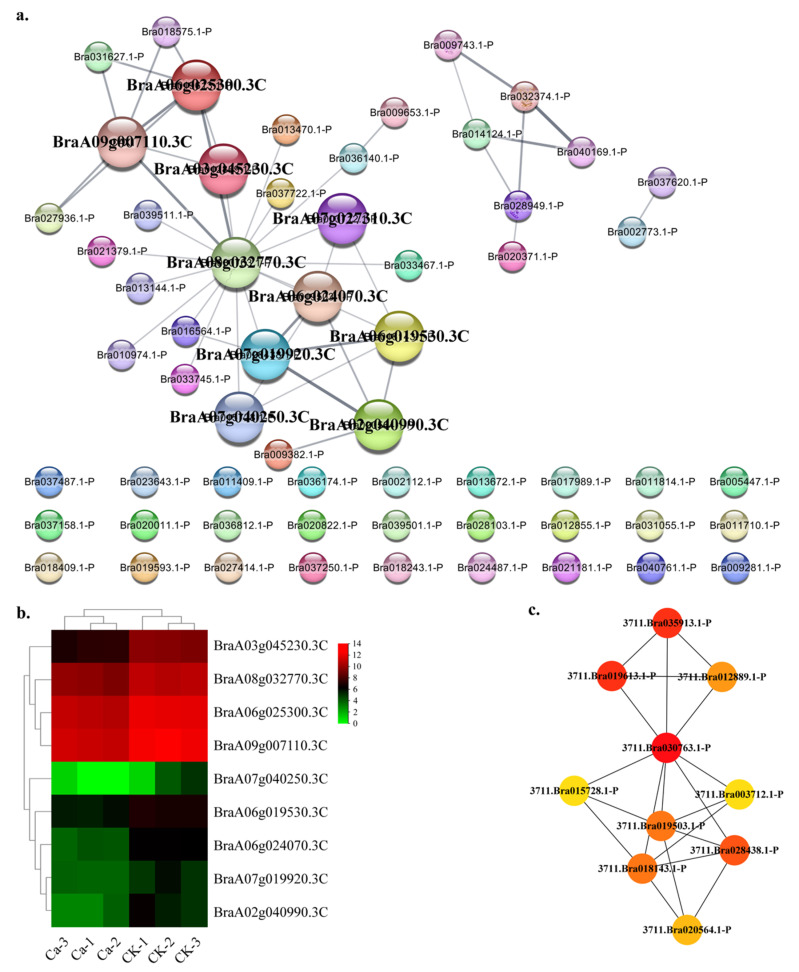
Protein–protein interaction (PPI) analysis in the Ca-related DEGs category. (**a**) PPI network in the Ca category. (**b**) Heatmap of the hub gene expression pattern. (**c**) Hub genes’ interactive relationship of the gene regulatory network.

**Figure 9 plants-11-03555-f009:**
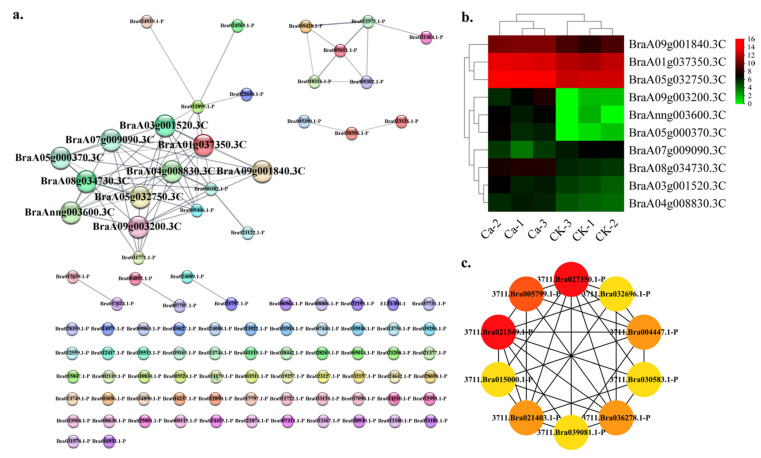
Protein–protein interaction (PPI) analysis in cell wall-related DEGs category. (**a**) PPI network in cell wall category. (**b**) Heatmap of the hub gene expression pattern. (**c**) Hub genes’ interactive relationship of the gene regulatory network.

**Figure 10 plants-11-03555-f010:**
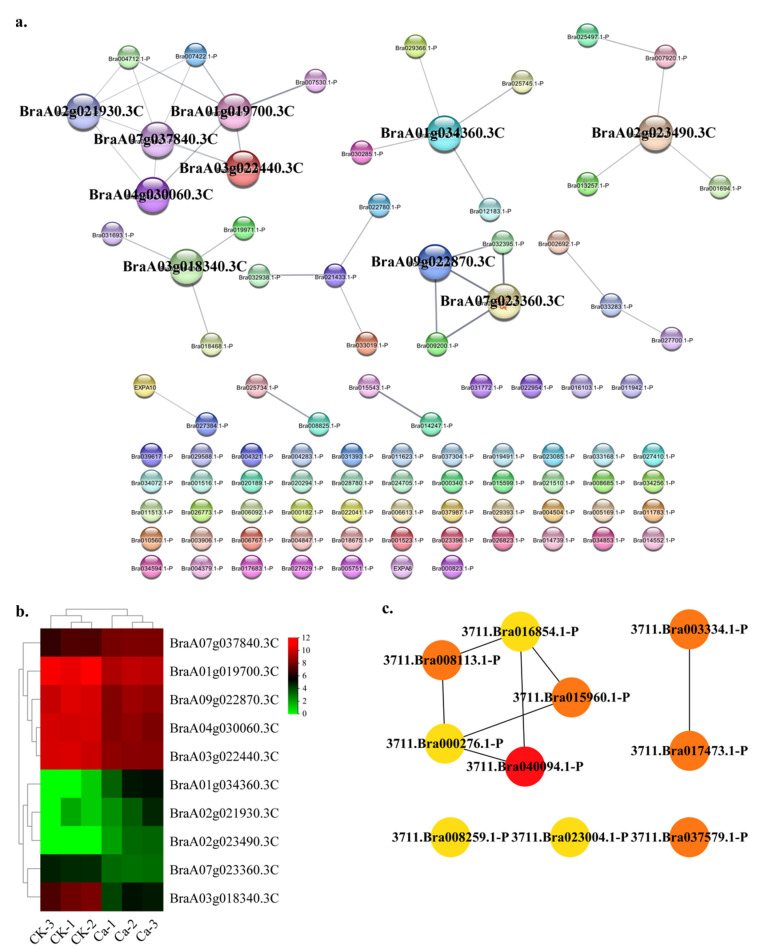
Protein–protein interaction (PPI) analysis in development-related DEGs category. (**a**) PPI network in the development category. (**b**) Heatmap of the hub gene expression pattern. (**c**) Hub genes’ interactive relationship of the gene regulatory network.

**Figure 11 plants-11-03555-f011:**
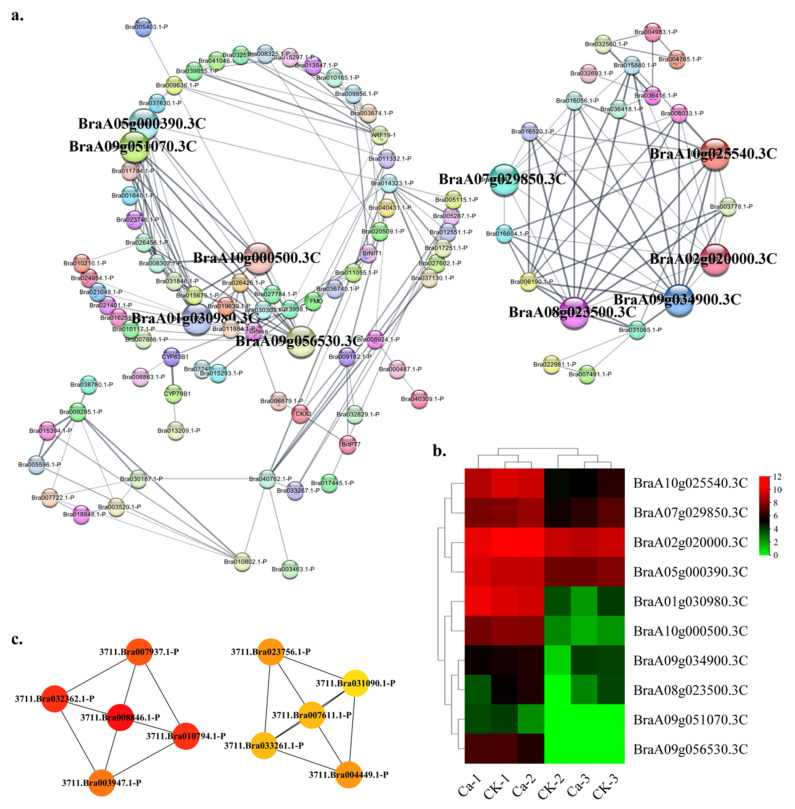
Protein–protein interaction (PPI) analysis in hormone-related DEGs category. (**a**) PPI network in the hormone category. (**b**) Heatmap of the hub gene expression pattern. (**c**) Hub genes’ interactive relationship of the gene regulatory network.

**Figure 12 plants-11-03555-f012:**
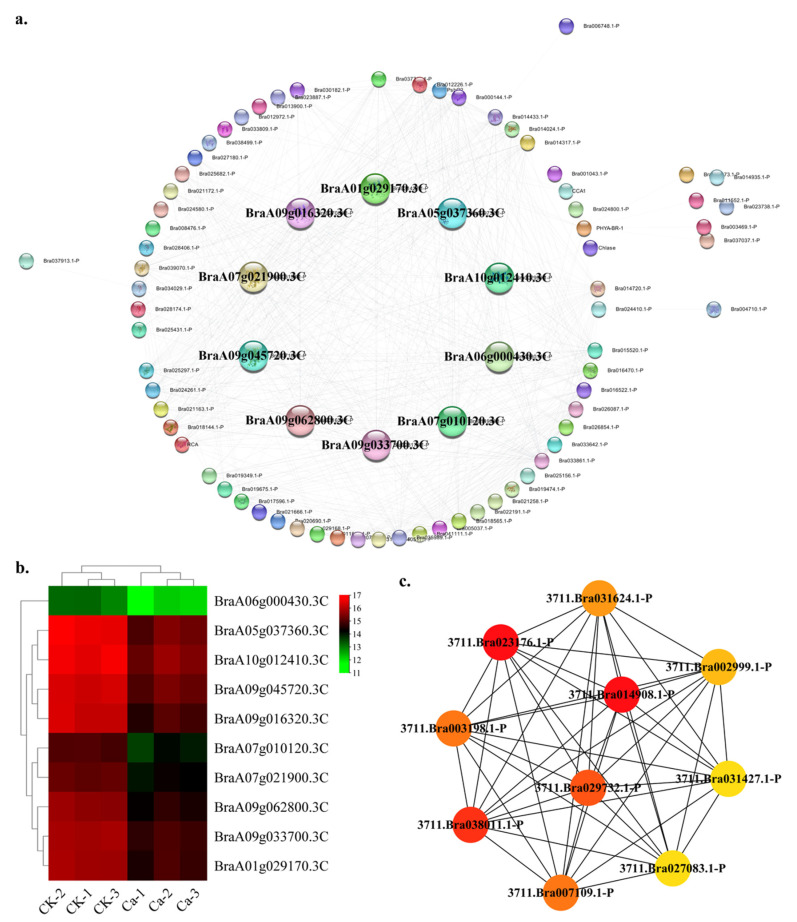
Protein–protein interaction (PPI) analysis in light-related DEGs category. (**a**) PPI network in light category. (**b**) Heatmap of the hub gene expression pattern. (**c**) Hub genes’ interactive relationship of the gene regulatory network.

**Figure 13 plants-11-03555-f013:**
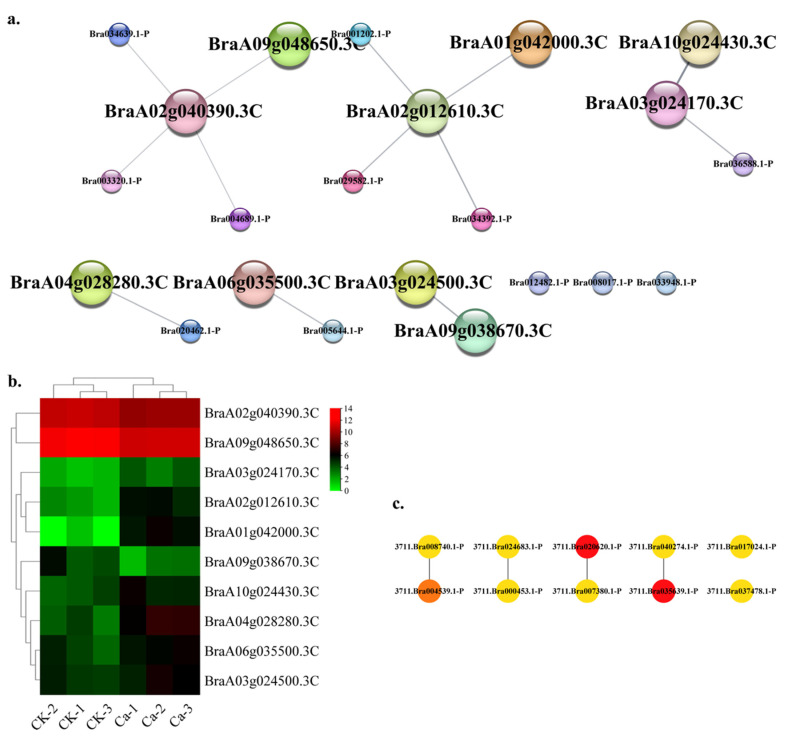
Protein–protein interaction (PPI) analysis in transcription factor (TF)-related DEGs category. (**a**) PPI network in the TF category. (**b**) Heatmap of the hub gene expression pattern. (**c**) Hub genes’ interactive relationship of the gene regulatory network.

**Figure 14 plants-11-03555-f014:**
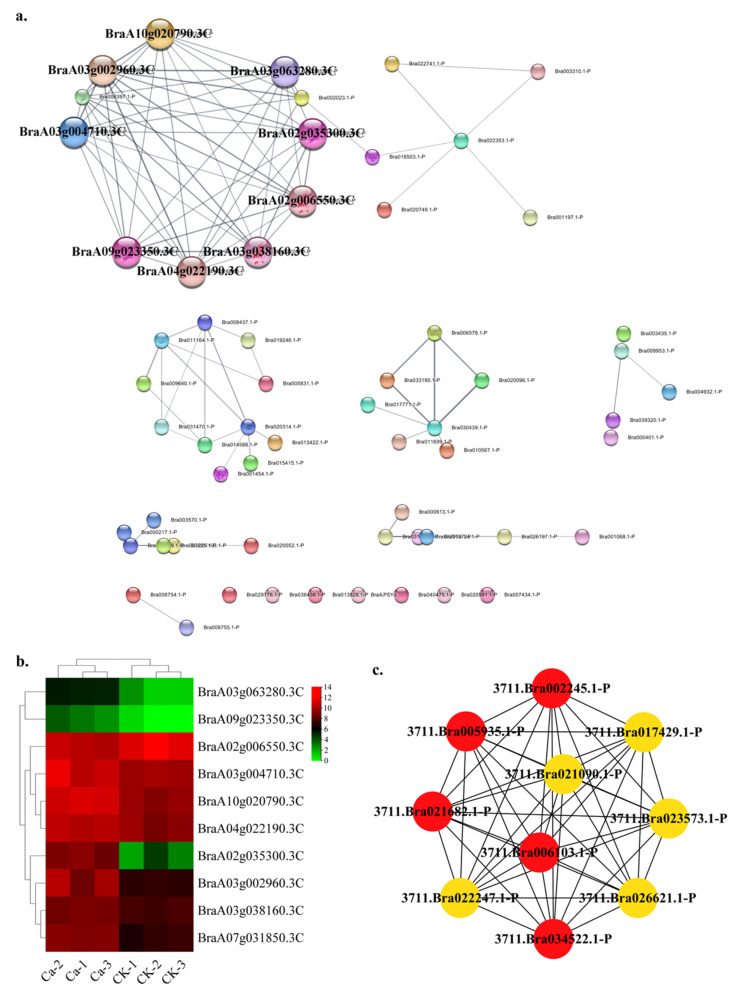
Protein–protein interaction (PPI) analysis in transporter-related DEGs category. (**a**) PPI network in the transporter category. (**b**) Heatmap of the hub gene expression pattern. (**c**) Hub genes’ interactive relationship of the gene regulatory network.

**Figure 15 plants-11-03555-f015:**
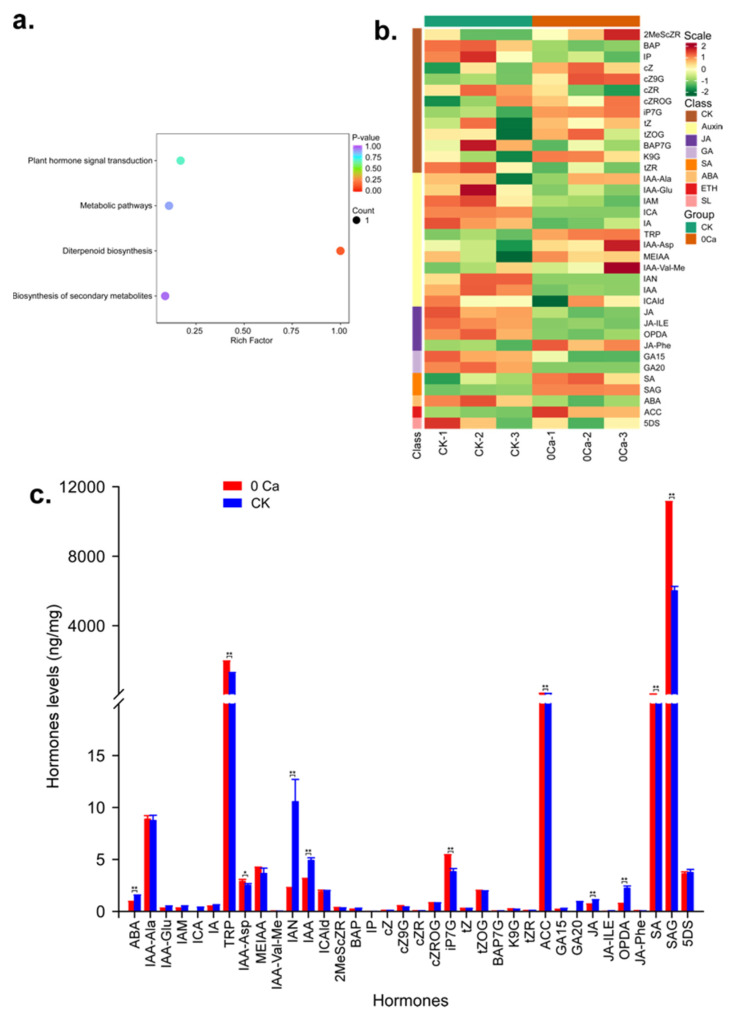
Hormone identification for metabolomic analysis. (**a**) KEGG enrichment of the DEMs. (**b**) Sample clustering diagram. (**c**) Histogram of hormone levels’ comparison between 0Ca and CK. Asterisks indicate statistical significance using Student’s *t*-test: *, *p* ≤ 0.05, **, *p* ≤ 0.01.

## Data Availability

The data supporting this study’s findings are available from the corresponding author, Jian-Wei Gao, upon reasonable request.

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
