# Peer review of "Comparative Transcriptome and Co-Expression Network Analyses Reveal the Molecular Mechanism of Calcium-Deficiency-Triggered Tipburn in Chinese Cabbage (Brassica rapa L. ssp. Pekinensis)"

_plants, 2022, doi:10.3390/plants11243555_

Round 1

Reviewer 1 Report

The research manuscript is titled " Comparative transcriptome and co-expression network analyses reveal the molecular mechanism of calcium-deficiency triggered tipburn in Chinese cabbage (Brassica rapa L. ssp. Pekinensis)”

The Introduction states the objectives of the work and provides an adequate background. The results, methods, and conclusion sections are well written. Overall, a good research article. However, this manuscript requires minor revision in certain sections.

1.     It might be useful to insert a flow chart in which it is explained how the study was built.

2.     Results section: Figure 6; please add jitters in the bar plots and add significant value for each DEG. Also, add the raw data in Table S4 and mention that in the results text.

3.     Please mention which type of PPI network did you extract from the STRING database? Was it all/ experimental/ or others (method section 4.5).

4.     Please add a limitations section of the study.

5.     Please update the author contribution section.

Reviewer 2 Report

The manuscript by Zhang et al discusses the effect of Ca deficiency on Cabbage. Despite not being the most novel topic, the manuscript provides enough new information to be of interest for the scientific community. Therefore, I think that it can be suitable for publication in Plants after few minor points have been addressed.

Authors use EGTA to chelate Ca from the growth medium, thus achieving Ca deficiency. Even though plants do display the typical Ca-deficient phenotype, authors should provide proof, via a citation or their own data, that this method is working as intended. That is, that only Ca is being chelated and plants are not suffering from the combined deficiency of several mineral elements.

The methodology regarding plant cultivation and sampling needs to be better described. Please, proved a citation or the composition of the employed Hoagland solid medium. Describe the type and dimensions of the pots or dishes in which plants were grown. What was the light intensity and humidity in the climate chamber? When authors write that leaves were collected on the 21st day, do they mean from sowing or from emergence? Were all the leaves of each plant collected, or only selected ones/?

Certain parts of the Results section are not pure description of the data, but more of a discussion of the results, and thus should be moved to the Discussion section. For example, Lines 238 to 249, L264-270, and L312-318.

The paragraph from line 188 to 214 is just a list of the amount of DEGs up- and down-regulated in each KEGG pathway and is therefore quite tedious. I would suggest including the numbers of regulated genes directly on the Figure 5b (or using a stacked bar graph) and deleting the whole paragraph; or at least limiting it to the most relevant pathways that authors would like to highlight.

Some other minor comments:

Figure 1. Please, include the age of the plants in the figure caption.

Figure 15. Control plants are labelled WT instead of CK. Try to keep the nomenclature homogeneous.

Figure 15c. Something does not seem correct with the split Y axis. According to the figure, it seems that TRP, ACC and SA have a concentration above 2000 ng/mL, but the Table S9 shows concentrations lower than that. Please, check.

The units displayed in Figure 15c are ng/mL. I suggest changing to the much more commonly used ng/mg fresh weight.

Most heatmaps included in the manuscript are not very colourblind-friendly. I suggest changing the colour-coding to the same one used in Figure 4 (blue-red).

The Table S9 contains Chinese (I think) characters.

L473. Please provide the citation for Amor et al.

L475. Please provide the citation for Rangnekar et al.

L491. Correct to “plant hormone”

L661. Please, provide the composition of the internal standard.

Reviewer 3 Report

The authors presented a good approach in the manuscript entitled Comparative Transcriptome and Co-expression Network Analyses Reveal the Molecular Mechanism of Calcium-Deficiency Triggered Tipburn in Chinese Cabbage (Brassica rapa L. Ssp. Pekinensis)”. This study analyses Chinese cabbage tipburn using RNA-seq data collected from calcium-deficient material. The authors discovered high-resolution coexpressed gene groups and complex biological pathways. The authors noted that molecular genetic methods, deduced GRNs, and transcriptional profiles can help discover promising candidate genes and determine their importance in Chinese cabbage tipburn. Before recommending this article for publication, some shortcomings should be resolved.

Authors have to provide details cohesively and logically. The authors should consider elaborating and be specific on the future scenarios of one of the important resources.

The authors should improve the grammar, spelling, punctuation, and overall English of the manuscript. 

The scientific names of the species and the names of the genes must be italicized in the manuscript. The abbreviations should be fully explained during the first mention in the abstract and introduction. 

1.     Please write more about Brassica rapa L. ssp. Pekinensis, please indicate the genus and family name 

2.     Please give full and proper legends in Figure 1 and Figure 6

3.     References should be according to journal guidelines.

4.  In material and methods, “Data sources and detection of conserved structure” should be shortened as the reader might lose pace while going through it. Instead, the author can mention a link for accession numbers reference.

TRANSLATE with x English
Arabic Hebrew Polish
Bulgarian Hindi Portuguese
Catalan Hmong Daw Romanian
Chinese Simplified Hungarian Russian
Chinese Traditional Indonesian Slovak
Czech Italian Slovenian
Danish Japanese Spanish
Dutch Klingon Swedish
English Korean Thai
Estonian Latvian Turkish
Finnish Lithuanian Ukrainian
French Malay Urdu
German Maltese Vietnamese
Greek Norwegian Welsh
Haitian Creole Persian  
TRANSLATE with COPY THE URL BELOW Back EMBED THE SNIPPET BELOW IN YOUR SITE Enable collaborative features and customize widget: Bing Webmaster Portal Back

Round 2

Reviewer 3 Report

The revised version of this manuscript can be accepted.

TRANSLATE with x English
Arabic Hebrew Polish
Bulgarian Hindi Portuguese
Catalan Hmong Daw Romanian
Chinese Simplified Hungarian Russian
Chinese Traditional Indonesian Slovak
Czech Italian Slovenian
Danish Japanese Spanish
Dutch Klingon Swedish
English Korean Thai
Estonian Latvian Turkish
Finnish Lithuanian Ukrainian
French Malay Urdu
German Maltese Vietnamese
Greek Norwegian Welsh
Haitian Creole Persian  
TRANSLATE with COPY THE URL BELOW Back EMBED THE SNIPPET BELOW IN YOUR SITE Enable collaborative features and customize widget: Bing Webmaster Portal Back